# Phantom rivers filter birds and bats by acoustic niche

D. G. E. Gomes [1✉], C. A. Toth[1], H. J. Cole[1], C. D. Francis [2,3] & J. R. Barber [1,3✉]

Natural sensory environments, despite strong potential for structuring systems, have been neglected in ecological theory. Here, we test the hypothesis that intense natural acoustic environments shape animal distributions and behavior by broadcasting whitewater river noise in montane riparian zones for two summers. Additionally, we use spectrally-altered river noise to explicitly test the effects of masking as a mechanism driving patterns. Using data from abundance and activity surveys across 60 locations, over two full breeding seasons, we find that both birds and bats avoid areas with high sound levels, while birds avoid frequencies that overlap with birdsong, and bats avoid higher frequencies more generally. We place 720 clay caterpillars in willows, and find that intense sound levels decrease foraging behavior in birds. For bats, we deploy foraging tests across 144 nights, consisting of robotic insect-wing mimics, and speakers broadcasting bat prey sounds, and find that bats appear to switch hunting strategies from passive listening to aerial hawking as sound levels increase. Natural acoustic environments are an underappreciated niche axis, a conclusion that serves to escalate the urgency of mitigating human-created noise.

[1] Boise State University, Boise, ID, USA. [2] California Polytechnic State University, San Luis Obispo, CA, USA. [3]These authors jointly supervised this work: C. D. Francis, J. R. Barber. ✉email: dylangomes@u.boisestate.edu; jessebarber@boisestate.edu

Animals surveil the environment to extract information important for decision making. Bats alter roost emergence in the presence of rain noise[1] and bees use ultraviolet[2] and electromagnetic[3] signals created by flowers to learn the location of nectar rewards. The information an animal can extract from the world, its *umwelt*[4], has long been appreciated as important for explaining animal behavior[5], yet we often fail to account for the filtering effects of these *umwelten* when explaining larger ecological patterns. Despite recent advances in understanding the role that anthropogenic noise and artificial light play on wildlife[6–10], it is surprising that natural sensory environments, such as gradients of light and sound, are rarely used in ecological analyses[11]. In one of the few exceptions, noise from nearby streams had the most power in explaining where frogs chose to call relative to other habitat variables[12]. Recent experimental evidence further supports a potentially widespread role of sensory environments in shaping animal behavior and ecology. Playback of river noise alters spider abundance[13], healthy coral reef sounds increases fish settlement on degraded reefs[14], and stream noise, paired with male advertisement calls, attracts more female torrent frogs[15].

The cacophony of an insect chorus and the thunder of a mountain river are examples of intense acoustic sources that characterize many environments. There are 150,000 km of marine shoreline (NOAA 2014) and 5.6 million km of rivers and streams in the United States alone (US EPA 2014) that expose adjacent environments to the sounds of moving water. We hypothesize that such intense natural acoustic sources have the power to structure habitat use[11]. To test this hypothesis, we select 60 locations within 20 sites, which we match for elevation and riparian vegetation, along streams in the Pioneer mountain range of Idaho (Fig. 1A) and monitor two taxonomic groups dependent upon the acoustic environment[5] that are abundant, diverse, and widespread across our system—birds and bats. Ten sites remain acoustically unaltered (controls) and span a natural range of sound levels; from quiet, slow-moving streams to loud, whitewater rapids (30.6–73.8 dBA, 24-h L50). We broadcast whitewater river noise from speaker arrays powered by solar panels and banks of batteries at five additional, naturally quiet streams using acoustic recordings taken from the highest sound level control sites. These phantom rivers thus present the amplitude and spectral profiles of raging, whitewater rapids (avg. median frequency ± SD: 2.1 ± 1.3 kHz). To understand the mechanisms underlying responses to the acoustic environment, we also create a gradient of background spectra by broadcasting shifted river noise of an identical temporal profile, but shifted upwards in frequency (4.8 ± 1.3 kHz) at five additional quiet-stream sites (Fig. 1B). We create these files so that the average broadcast energy, weighted by birds' hearing thresholds, is the same (see supplement for details; Fig. 1C).

Energetic masking occurs when there is spectral overlap between the signal and background noise. Masking of vocalizations, like birdsong, can drive distributional shifts of animals in areas exposed to anthropogenic noise[16]. Similarly, the masking of prey cues is suggested to be a primary mechanism structuring the space use of acoustically-mediated predators, such as gleaning bats[17,18].

In this work, we predict that overlap between song and background noise is an important predictor of bird distributions if masking of birdsong is underlying noise effects (Fig. 1D). Because most bat echolocation is higher frequency than the acoustic environments we created, we do not expect changes in bat activity to be related to sonar frequency, yet we do predict that gleaning bats will avoid sites with energy in higher frequencies (>3 kHz)[19] due to masking of prey-generated sounds (Fig. S11)[17,18]. Our experimental design allows us to explicitly test the effects of sound level separately from those of background spectra.

## Results and discussion

**Bird abundance.** Leveraging data from 2969 point counts (~150 count hours), we found bird abundance declined by 7.0% (95% CI: 3.4–10.5%) for each 12 dB increase in sound level (Fig. 2A; Table S1). High-intensity noise makes detection and discrimination of acoustic signals and cues more difficult, either because of energetic masking at the periphery of the auditory system, or because of limited central attentional resources[6]. To explore masking of communication, we took the difference between the median background frequency and individual bird species' peak vocalization frequency as a measure of spectral overlap with the acoustic environment. Birds with a peak vocalization frequency closer to the median of the background spectrum showed lower abundances, with declines of 10.0% (5.1–15.3%) for each 2 kHz increase in spectral overlap (Fig. 2B). However, these overlap-mediated effects interact with sound level in a diminishing way (Table S1): higher amplitude background noise resulted in weaker relationships between spectral overlap and bird abundance. It seems that when acoustic environments are intense, masking of specific vocalizations is no longer the primary mechanism underlying distributional changes (Fig. S14).

Individual species models (Table S2) combined with phylogenetically-informed, trait-based analyses indicate that birds with lower-frequency songs avoid noise with similar spectra, while birds with higher frequency vocalizations do not ($t = -3.73$; $p < 0.01$; Fig. 2C). Previous work (2011)[20] found that lower-frequency vocalizers more strongly avoid high sound levels. Here, no distributional patterns related to song frequency emerged in response to the sound level or median frequency of the acoustic environment (Table S3).

**Bird foraging.** Animals that remain in anthropogenic noise can bear costs, such as reduced body condition[6]. To examine one potential behavioral cost of exposure to natural noise, we placed 720 clay caterpillars across our sites (Fig. S8). While controlling for bird abundance, the odds of a caterpillar being depredated by a bird decreased by 37.2% (95% CI: 22.7–49.1%) for each 12 dB increase in sound level (Fig. 2D; Table S4). As this task was entirely visual, it seems likely that cross-modal attentional limitations underlie this effect[21]. Birds that persist in high sound-level environments will likely suffer negative foraging consequences under noise exposure and such effects may have indirect consequences for arthropods[13].

**Bat activity.** The direct effects of the acoustic environment are a potential driver of bat distributions. Limited evidence suggests that space-use by bats is shaped by anthropogenic noise[22], and laboratory work has shown that gleaning bats have difficulty hunting in both anthropogenic and natural noise[17,18]. Using ~100,000 identified bat call sequences, we found that overall bat activity decreased 8.2% (95% CI: 4.8–11.4%) for each 12 dB increase in sound level, and decreased 19.5% (16.1–22.8%) for each 2 kHz increase in median background frequency (Fig. 3A, B; Table S5). Individual species models reveal consistently similar inferences (Table S6). Bats likely perceive higher frequency noise as louder[23], yet masking of echolocation is an improbable explanation for these results as bat sonar does not spectrally overlap with the acoustic environments we studied (although see Bunkley et al.[22] for frequency shifts in non-overlapping noise). Phylogenetically controlled trait-based analyses revealed that bats with increasingly high-frequency sonar exhibited increasingly lower activity with rising sound levels ($t = -5.39$; $p < 0.001$; Fig. 3C; Table S8), further counter to masking as an explanatory mechanism. This finding may reflect indirect drivers if small insects disproportionately avoid noise,

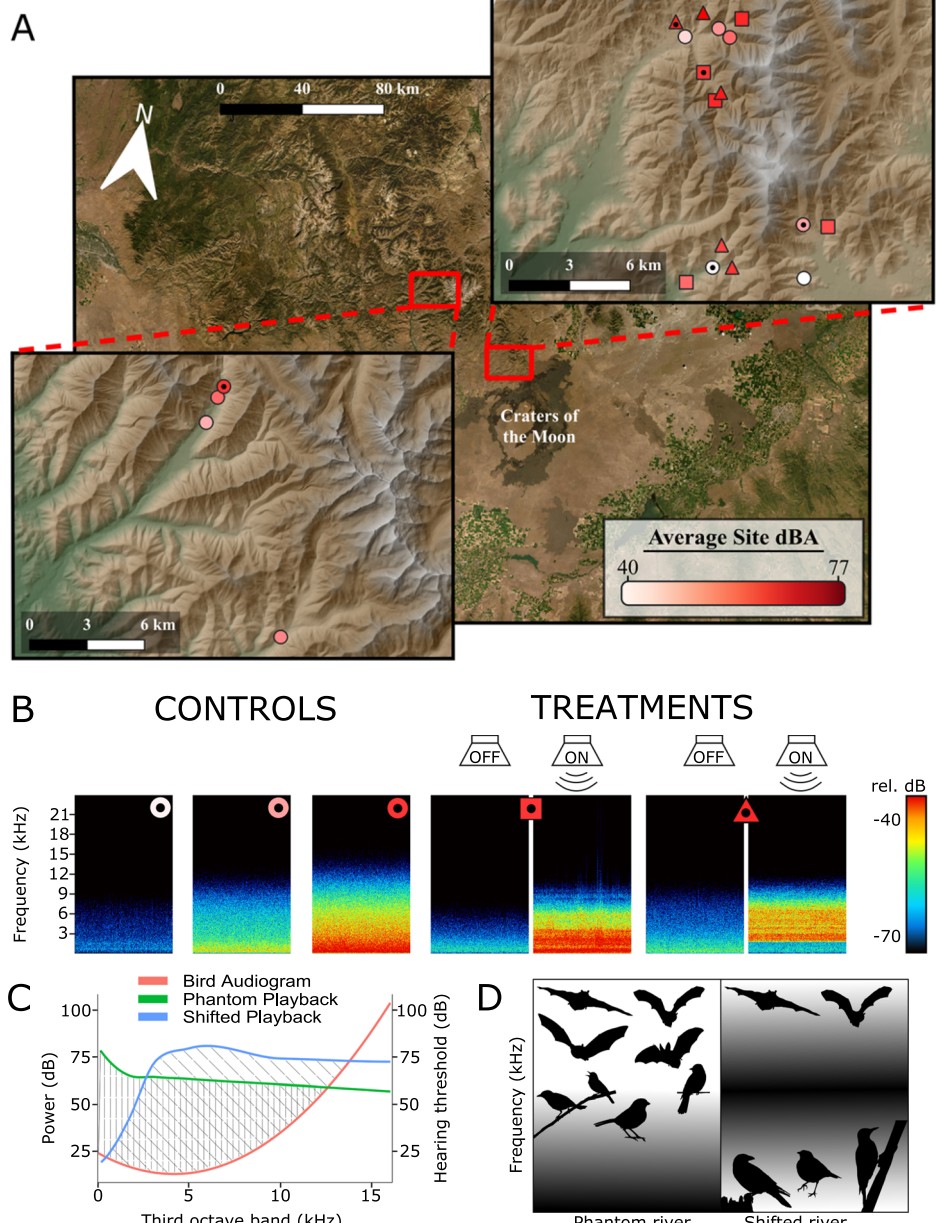

**Fig. 1 Design and predictions for phantom rivers playback experiment. A** Twenty sites were monitored across the Pioneer Mountains of Idaho, comprising a gradient of sound levels (indicated by color scale). Control sites are indicated by circles, phantom river playback sites by squares, and shifted-river playback sites by triangles. **B** These symbols are matched with their geographically referenced representative spectrograms (linked between **A**, **B** via black dots in center of symbols) to show the gradient of noise exposure at control sites and playback sites with speakers both on and off. **C** Both playback files were created so that the average broadcast energy, weighted by birds' hearing thresholds, was the same. **D** Predicted responses of birds and bats to the playback treatments. Silhouettes placed on frequency axis as a heuristic representation of vocalization frequency (not to scale).

as these are most accessible to high-frequency echolocators[24]. Alternatively, high-frequency echolocators (and listeners[23]) experience a reduced sensing area, since high-frequencies attenuate quickly, which may compromise risk assessments in a noisy world.

While these data suggest that masking of echolocation is not responsible for patterns of bat activity, they do indicate that limited attentional resources available for sonar processing and perhaps masking of lower-frequency environmental cues[25] might be two underappreciated drivers of bat distributions. We used additional trait-based analyses to test a component of the latter hypothesis and found that bats capable of hunting via passive listening are not more likely to avoid noise (Table S8). This result

is at odds with previous laboratory work[17], and might indicate that wild bats are behaviorally flexible enough to cope with noise[18]. Indeed, those bats that are obligate aerial hawkers are more likely to avoid higher frequency acoustic environments ($t = -4.1$; $p < 0.01$).

**Bat foraging**. To quantify bats' use of passive listening and active sonar strategies, and to explore if bats employ flexibility in hunting techniques to circumvent the costs of noise (Fig. S11), we deployed custom-designed assays at 36 locations across our sites. We placed small speakers playing insect walking and orthopteran mating sounds on the ground to evaluate bats'

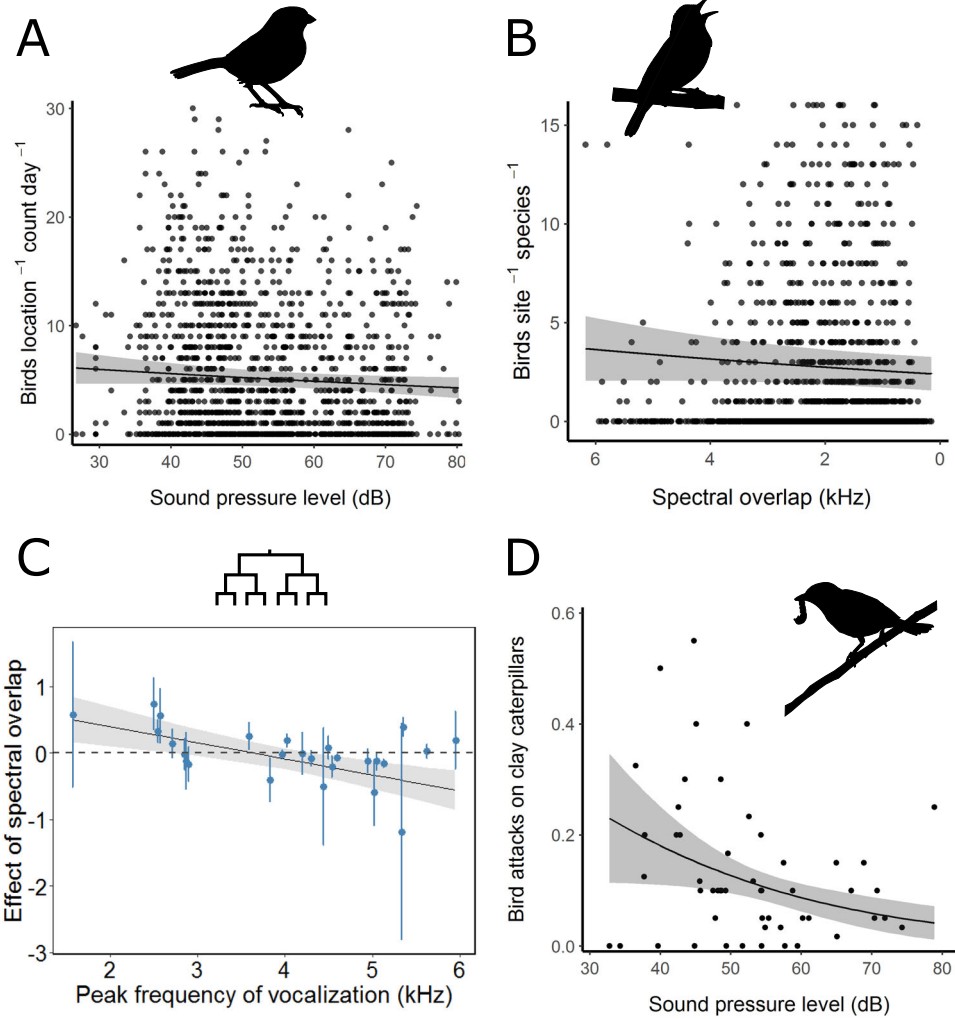

**Fig. 2 Bird responses to noise. A** Bird abundance declines with increasing sound levels. **B** Bird abundance declines with increasing spectral overlap with song (note reversed x axis; 0 = complete spectral overlap). **C** Phylogenetically-informed trait analyses reveal that lower-frequency singers are more likely to be masked by background noise with similar spectra (n = 26 species). **D** Bird foraging rates decrease in high sound levels. For all plots, points represent raw data (species estimates for **C**), error bars represent standard errors (**C** only), the plotted lines (estimated conditional means) and shaded gray regions (95% confidence intervals) represent predicted values of the response variable over the range of the variable on the x axis, given constant mean values of all other variables in the model.

attraction to prey-generated cues (Fig. S13). To query bats' use of sonar-mediated aerial hawking, we used a motor-driven synthetic wing[sensu 26] placed at 1 m above the ground to echo-acoustically mimic the wingbeat frequency of insect prey (250 Hz, e.g., Diptera[27]; Fig. S12). We focused our analysis on bat species that employ both strategies (i.e., behaviorally flexible bats; including *Corynorhinus townsendii*, *Myotis evotis*, *Myotis lucifugus*, and *Myotis thysanodes*; see Supplementary Information) and predicted that high sound-level acoustic environments would hinder bats' use of passive listening (which requires the processing of lower-frequency sounds) and result in heightened use of sonar[18,26]. Indeed, while controlling for changes in bat abundance, for every 12 dB increase in sound pressure level and 2 kHz increase in median background frequency, bat activity at speakers playing prey cues decreased by a factor of 0.58 (95% CI: 0.38–0.87; Fig. 3D; Table S7), while aerial hawking activity increased by a factor of 8.1 (95% CI: 1.5–44.1) at simultaneously deployed robo-insects. This strategy switching only seemed to occur at sites with relatively higher frequency acoustic environments, likely because the bulk of the energy of prey-generated sounds are within these frequencies[25]. The ability to behaviorally

switch is unlikely to be universal, which may allow flexible bat species to persist where others cannot[18].

**Detectability in noise**. We experimentally show that natural noise can have strong effects on animal abundance, activity, and behavior, yet our findings are dependent on the probability of detecting vocalizing animals in noise[28,29]. For birds, we controlled for this potential problem using four approaches. We turned off speakers during counts so that most observations occurred below sound levels known to interfere with detection[28] (Fig. S6). To implement imperfect detection into our models, we both directly measured observer detection in noise with a birdsong playback experiment (Fig. S7), and estimated bird detection probabilities with a noise-informed removal model (Table S11). We wore earplugs and earmuffs during a duplicate set of point counts so that observations were visual-only (Table S12), which suggest similar inferences as above (Table S1). For bats, a laboratory test verified acoustic recording units were triggered similarly in a gradient of noise levels. Further, we validated that identification software correctly classified bats by experimentally adding noise to files (see Supplementary information).

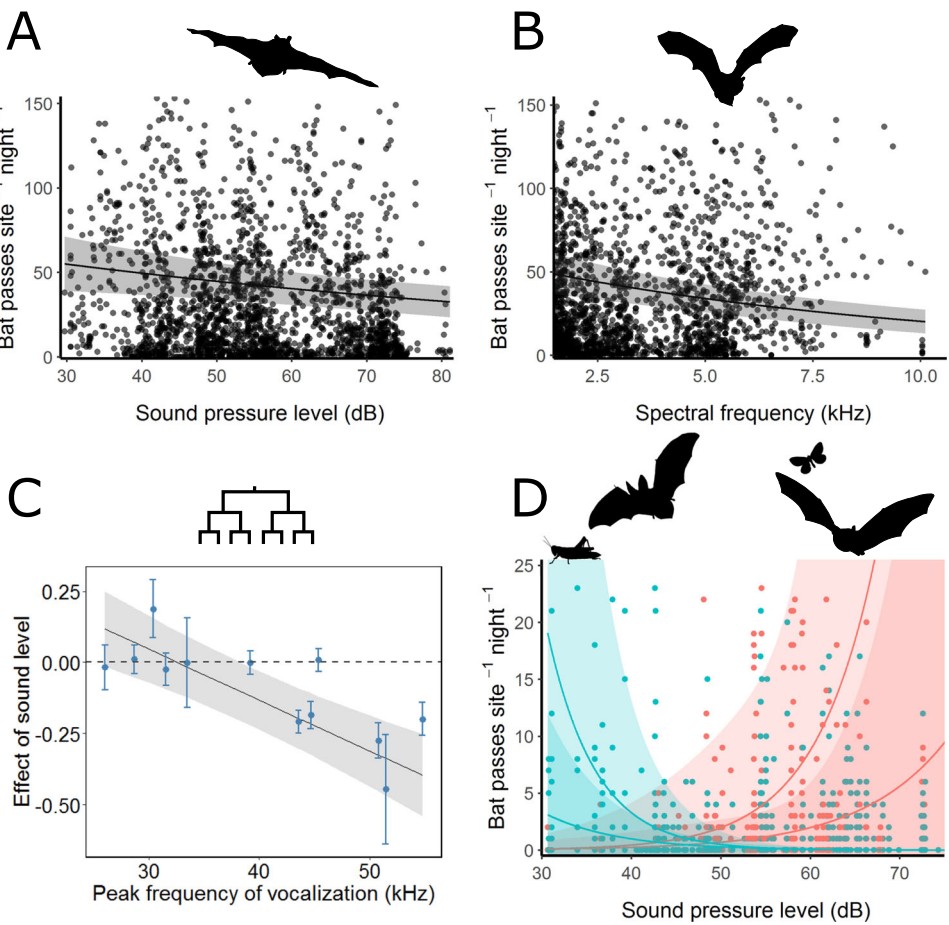

**Fig. 3 Bat responses to noise. A** Bat activity declines with increasing sound levels. **B** Bat activity declines with increasing frequency of the acoustic environment. **C** Phylogenetically-informed trait analyses reveal that higher frequency echolocators are more likely to avoid high sound-level environments ($n = 12$ species). **D** Foraging modality for flexible bat species appear to shift from passive listening to aerial-hawking with variation in background sound level and frequency. Blue lines denote predicted declines in bat detections at artificial prey sounds speakers with increased sound level where the top and bottom lines reflect median background frequencies at 10 and 8 kHz, respectively. Red lines denote the predicted increase in bat detections at robotic fluttering insects with higher noise levels at 8 kHz (top line) and 6 kHz (bottom line). For all plots, points represent raw data (species estimates for **C**), error bars represent standard errors (**C** only), the plotted lines (estimated conditional means) and shaded regions (95% confidence intervals) represent predicted values of the response variable over the range of the variable on the *x* axis, given constant mean values of all other variables in the model (except in **D**).

**Concluding remarks**. Our results demonstrate that natural acoustic environments represent an underappreciated dimension of the niche and are clearly important in shaping animal behavior and distributions. Incorporating this axis into our understanding of the natural world will provide stronger inference for both basic and applied questions[11]. Because the spatial and temporal footprint of human-generated noise is orders of magnitude greater than loud natural acoustic environments, it is critical to understand that the insights provided by our data increase the importance of mitigating noise pollution impacts on animals and their habitats. Our results reveal age-old strategies for dealing with the long-standing problem of noise and help explain contemporary responses to anthropogenic noise. A renewed focus on animal umwelten will redefine our understanding of ecological niche axes that have been canalized by our own sensory biases[11].

## Methods

IACUC approval: all work described below was approved by the Boise State Institutional Animal Care and Use Committee: AC15-021.

**Site layout**. We selected 20 sites, across five drainages, within the Pioneer Mountains of Idaho—matched for elevation and riparian habitat. We split these 20 sites into 10 noise playback sites, and 10 control sites (Fig. 1A; S1). The control sites ranged from quiet, slow-moving streams to relatively loud whitewater torrents. Noise playback sites, on the other hand, were relatively quiet (not whitewater) sites, where we broadcast loud whitewater river recordings with speaker arrays hung from towers (Fig. S1; S2; S3; S4; see supplementary information for more details on noise file creation, playback equipment, and experimental setup). At five of the noise playback sites we broadcast normal river noise (hereafter referred to as 'river noise' sites), and at the other five noise sites we broadcast spectrally-altered river recordings (hereafter referred to as "shifted noise" sites).

Our field sites were oriented along the riparian zone, with data collection occurring at three primary locations within each site (Fig. S1): (1) roughly in the middle of the speaker tower systems, (2) at a shorter distance from the middle location (mean: 198.2 ± 54.5 m SD; range: 117.6–384.5 m), and (3) and a longer distance from the middle location (in the opposite direction from the nearer location; mean: 312.7 ± 64.7 m SD; range: 249.1–479.6 m). Thus, sites were approximately 510.9 ± 98.3 m long (range: 374.7–850.6 m), along the riparian corridor. All control sites were, at minimum, 1 km apart along the riparian corridor from any noise site, to maintain acoustic independence (see Fig. 1A; S1).

### Data collection

#### Birds

We conducted three-minute avian point counts between one half hour before sunrise and 6 h after sunrise (roughly 0530–1130 h). During the project, we conducted 1330 point-counts from 28 May to 20 July 2017 and 1639 point-count events occurred from 7 May to 24 July in 2018.

*Caterpillar deployment.* We deployed a total of 720 clay caterpillars throughout the 2018 breeding season. Forty caterpillars were glued to stems and branches of trees between 1 and 2.5 m high at each site (Fig. S8). Twenty caterpillars surrounded the middle point count location at each site (a set of 10 were placed upstream, and another set of 10 were placed downstream starting from the middle ARU location), while the other twenty were at upstream and downstream sampling locations (10 each at upstream and downstream locations). We placed each caterpillar along the riparian corridor, at least 1 m apart from each other[30]. See Supplementary information for details on caterpillar predation scoring.

*Bird trait analysis.* We performed a trait-based analysis to understand the mechanistic patterns of bird distributions in our study paradigm. Avian vocal frequencies and body mass were collected from Hu and Cardoso 2009, Cardoso 2014, and Francis 2015[16,31,32]. When multiple sources contained data, the values were averaged. There were a few cases where none of those sources contained a vocal frequency or mass measurement for species of interest. Thus, representative songs were downloaded from the Macaulay Library of the Cornell Lab of Ornithology based on recording quality and geographical relevance (MacGillivray's warbler: ML42249; dusky flycatcher: ML534684; red-naped sapsuckers: ML6956), and analyzed with Avisoft SASLab Pro to obtain a peak frequency measure. Mass measurements for these 'missing' birds were taken from the 'All about birds' webpage of the Cornell Lab of Ornithology.

## Bats

*Measuring and identifying bat calls.* We measured bat activity using Song Meter 3 (hereafter "SM3") recording units (Wildlife Acoustics Inc., Massachusetts, USA) equipped with a single SMU (Wildlife Acoustics Inc.) ultrasonic microphone. One recording unit was used at each site and we pseudo-randomly rotated the unit between the three point-count locations so that each location was monitored for at least 21 days. We mounted microphones on metal conduit at a height of ~3 m, oriented perpendicular to the ground and facing away from the stream to optimize recording conditions (Fig. S9; S10; see Supplementary information for more information).

*Robotic insects.* We used a modified version of Lazure and Fenton's[26] apparatus to present bats with a fluttering target (Fig. S12). This consisted of a 3 cm² piece of masking tape affixed to a metal rod [30.48 cm length × 3.25 mm diameter], which itself was connected to a 12-volt brushed DC motor (AndyMark 9015 12 V, AndyMark Inc., Kokomo, IN, USA). The no-load revolution speed of these motors (267 Hz) falls within the range of wingbeat frequency measured in Chironomidae[27,33], a group that is an important food source for many North American bat species[34].

We attached each motor to a tripod made of PVC piping and positioned the tripod such that the target was approximately 1.2 m above the ground. Each motor was powered by a 12 V battery (35Ah AGM; DURA12-35C, Duracell) which was controlled by a programmable 12 V timer (CN101, FAVOLCANO) to automatically start and stop the motor each night. The rotors were powered for 2 h following sunset.

*Prey-sound speaker playback.* We created a playlist composed of several insect acoustic cues to present gleaning bats: a beetle (*Tenebrio molitor*) walking on dried grass, a cricket (*Acheta domesticus*) walking on leaves, mealworm larvae (*Tenebrio molitor*) on leaves, fall field cricket (*Gryllus pennsylvanicus*) calls, and fork-tailed bush katydid (*Scudderia furcata*) calls. The cricket and katydid calls were sourced from the Macaulay Library (ML527360 and ML107505, respectively).

*Experimental setup for bat foraging tests.* Most sites received two rotors (Fig. S12) and two speakers (Fig. S13): one of each at the center of the site, and one of each at approximately 125 m from the center of the site (in opposite directions in order to have tests in a range of acoustic environments), placed roughly 10 m from the edge of the riparian zone. Rotors and speakers at the center locations were separated by at least 50 m. The exception to this setup were the four positive control (loud whitewater river) sites, which only received a single rotor and speaker separated by 50 m because of logistical difficulties of accessing those sites. We paired each rotor and speaker with an SM2BAT + bat detector equipped with an SMX-US microphone (Wildlife Acoustics Inc.)[35], using tripods to elevate the microphones approximately 1 m off the ground and ~1 m from the speaker/rotor. We programmed the bat detectors with a gain of 36 dB and a trigger level of 18 dB to limit recordings to bats that were passing within the immediate vicinity. To allow for a comparison of activity between speakers and rotors, bat activity was only considered for the first two hours following sunset.

*Bat trait analysis.* We collected bat foraging behavior and peak echolocation frequency information to use as predictors in a phylogenetically controlled trait analysis (Tables S8; S13). We based our behavioral foraging classifications on the categories of Ratcliffe et al.[36] and followed the classifications of Gordon et al.[37] where possible, and others[38–43] where necessary. We extracted peak echolocation frequency from the 2017 and 2018 SM3 field recordings and employed two controls to decrease variability in call parameters potentially introduced via this method.

First, we selected only recordings made on control sites in 2017 and 2018 ($n = 740,848$ calls), as echolocation call characteristics may be affected by local acoustic environments (e.g., Bunkley et al.)[22]. Secondly, we averaged all call parameters per species per hour at each site to decrease the possible effects of few individuals driving measurements. This resulted in 9538 species-hours of recordings, which themselves were averaged per species (Table S13).

**Quantifying environmental variables.** We used long-term monitoring of the acoustic environment (via Roland R05 recorders) to calculate daily sound pressure level (L50 dBA) and median frequency (kHz) values for each location (see supplementary information for details on quantification of all predictor variables).

*Sound pressure level (SPL).* We converted 106,769 h of long-term ARU recordings into daily-averaged median sound pressure levels (L50; measured as dBA rel. 20 µPa) see refs. [13,44] using custom software 'AUDIO2NVSPL' and 'Acoustic Monitoring Toolbox' (Damon Joyce, Natural Sounds and Night Skies Division, National Park Service).

*Acoustic environment spectrum.* We used custom software[45] in the programming language R and the package 'FFmpeg' in command prompt to convert 106,769 h of long-term recordings into 71,282 individual 3-minute files starting each hour of the day (Fig. S5). Thus 24, 3-min files were created per acoustic recording location per day (one for every hour). We then used the packages "tuneR" and "seewave" to read in and measure the median frequency of sound files, respectively[45–47]. These hourly metrics were then averaged by date to create a daily metric.

**Statistics.** All models of abundance, activity, and foraging transects were generalized linear mixed effects models (glmm) in R[48] using the package 'lme4'[49,50] or 'glmmTMB'[51]. All distribution families were selected based on theoretical sampling processes of the data, models were checked for collinearity (VIF scores)[52], and model fits were visually checked with residual plots (see supplemental R code)[53].

*Bird abundance and bat activity*

Model predictors and covariates
Both bird and bat models had the following variables in a glmm: site and bird/bat species were random effects terms and sound pressure level (dBA L50), sound spectrum (median frequency), the interaction between sound pressure level and spectrum, elevation, percent riparian vegetation, ordinal date (and a quadratic version of this), and year as fixed effects. While year is sometimes used as a random-effect term, it is suggested to be used as a fixed effect if fewer than five levels exist for that factor, as variance estimates become imprecise[54,55]. Additionally, moon phase was a fixed effect in the bat models[56], while spectral overlap (the absolute difference between sound spectrum and bird species vocalization frequencies) and the interaction between sound pressure level and spectral overlap were fixed effects in bird models.

We attempted to fit both sound pressure level and spectrum as having random slopes for each species, yet both bat and bird models would not converge with such complex model structure. Thus, we followed group models with individual species models (see Supplementary information).

Model family distribution and link function
For both bird and bat counts, we used a negative binomial distribution with a log link, rather than a Poisson distribution, because data were over-dispersed. We plotted variance-mean relationships and residuals of multiple models to select the appropriate variance structure, and compared these with AIC to select the best-fitting distribution (see R script for further justification of these methods)[54].

Individual species models
Individual species models were parameterized the same as above (except without the species term). All 12 bat species (see Tables S6; S10) and 26 of the most common birds (see Tables S2; S9) were modeled individually to be able to interpret model parameter estimates, with complex interactions, for each species.

*Clay caterpillar predation.* We modeled caterpillar predation with a glmm (binomial family; logit link function), using the number of individual scorers as weights in the model. Like the bird abundance model, we used site as a random effect and sound pressure level (dBA L50), spectral frequency (median), elevation, percent riparian vegetation, ordinal date, and year as fixed effects (Table S4). Additionally, the predicted number of birds at a site were modeled as fixed effects to control for varying amounts of foraging birds on the landscape.

*Robotic moths and prey-sound speakers.* Robotic moth and prey-sound speaker models were parameterized exactly the same as the overall bat activity model. That is, the model was fit with a negative binomial family (log link) with site and species as random effects and sound pressure level (dBA L50), sound spectrum (median frequency), the interaction between sound pressure level and spectrum, moon phase, elevation, percent riparian vegetation, ordinal date (and a quadratic version of this), and year as fixed effects. Additionally, the predicted number of bats at a

site were modeled as fixed effects to control for varying amounts of foraging bats on the landscape.

*Trait analyses.* We performed trait analyses with phylogenetic generalized least squares (PGLS) to control for relatedness while predicting species responses to noise[12]. We performed PGLS analyses with the *gls* function in the R package nlme[57], and accounted for error in the response variable with a fixed-variance weighting function of one divided by the square root of the standard error of the response estimate[58,59]. We accounted for phylogenetic structure by estimating Pagel's λ[60]. When λ estimates fell outside of the zero to 1 range, we fixed λ at the nearest boundary. For bird models, we used a pruned consensus tree from a recent class-wide phylogeny[61]. For bats, we used a pruned mammalian tree[62]. We used initial global models with all traits as variables that explained the responses to sound pressure level (SPL; birds and bats), spectral overlap with birdsong (birds), background frequency (bats), and the interaction between SPL and each measure of frequency (birds and bats). We then used AIC model selection[63] to choose top models in explaining these patterns. Models with dAIC ≤4 are included in Table S3 (birds) and Table S8 (bats), and the top model is interpreted in the main text.

**Reporting summary**. Further information on research design is available in the Nature Research Reporting Summary linked to this article.

## Data availability
All source data are provided with this paper at Dryad Data Repository (https://doi.org/10.5061/dryad.n5tb2rbsv).

## Code availability
All code and materials used in the analysis are available as a fully-reproducible workflow from Dryad Data Repository (https://doi.org/10.5061/dryad.n5tb2rbsv).

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

## Acknowledgements

Special thanks to Brian and Kathleen Bean for allowing us access to their land, Lava Lake Ranch, where this research took place. We thank Elizeth Cinto-Mejia, Krystie Miner, Will McDonald, Laura Grace Barta, Ben Sweet, Kate Sweet, Nicholas Carlson, Yael Lehnardt, William Prum, Christine Petersen, Amanda Emmel, Charlotte Cumberworth, and Blair Boyt for their help with fieldwork, Juliette Rubin for assistance with detection of bat calls in noise, Ian Robertson, Trevor Caughlin, Matt Williamson, and the students of EEB 607 for helpful comments on early versions of this manuscript, Kurt Fristrup for frequent and valuable consultation, and Greg Carr for offering us a quiet place to write this paper. We thank NSF for funding (GRFP ID 2018268606 to D.G.E.G., DEB 1556192 to C.D.F., and DEB 1556177 and IOS 1920936 to J.R.B.).

## Author contributions

J.R.B. and C.D.F. conceived of the overall project design and designed the main experiments. J.R.B., D.G.E.G., and C.A.T. designed the bird and bat foraging experiments. J.R.B. supervised implementation of the experiment. D.G.E.G. coordinated and led the field team. D.G.E.G. and C.A.T. processed sound files. D.G.E.G., C.A.T., and H.J.C. collected data. C.A.T. and H.J.C. processed acoustic monitoring data. All authors interpreted data. D.G.E.G. led data analysis. C.D.F. and J.R.B. contributed to data analysis. D.G.E.G. and J.R.B. drafted the initial manuscript. D.G.E.G., C.A.T., C.D.F., and J.R.B. critically revised the manuscript. All authors approved of the final version of the manuscript.

## Competing interests

The authors declare no competing interests.
