## [Peer Review File · Nature Communications]

REVIEWER COMMENTS

Reviewer #1 (Remarks to the Author):

This is a great piece of work, containing an impressive amount of field observations and experimental manipulations. The authors compared natural sites with varying levels of riverstream noise with sites where they broadcasted the same as well as upshifted noise through speaker arrays for two seasons. The results are extensive and very interesting and will stimulate a lot of future research.

I have only a few comments/questions regarding the methods and results, which I have added to the manuscript. My main question is why the authors did not provide details on comparisons between the control and experimental sites and only report patterns with noise level or frequency content in their main text. Perhaps it's hidden in the supplementary files, but I thought it was a bit odd to provide a lot of details on the sampling setup which was not reflected in the results. I also have some concerns regarding the results plotted in Fig 2. The regression line and CI don't seem to match the reported effects in the text, so I would advise the authors to have an other careful look at the models or the figure.

Keep up the good work!

Reviewer #2 (Remarks to the Author):

Major claims: natural noise impacts the distribution of animals along the noise gradient and can alter their foraging behavior. >Novelty: NATURAL noise, gradient-impact quantification, behavioral changes even cross-sensory (none of them demonstrated before or only poorly substantiated)
Importance to community: authors show evolutionary basis of adaptations in occurrence and implications on their vocalizations, a topic of speculation but up to now severely lacking in data.
"Impact of natural acoustics on ecology" is a very general topic that few other studies contributed to as much as the current study.

This manuscript is a landmark contribution to the current knowledge of how noise shapes acoustic communities and how it affects local abundance of birds and bats. Much attention has focused on the impact of anthropogenic noise, even with and without covid-lockdowns, or on presumed acoustic adaptations by birds to anthropogenic noise (some of which are still being questioned). Many studies on the impact of anthropogenic noise have no valid control, simply because uninhabited areas with no roads (and noise) also tend to constitute more attractive habitats. The one and only scientifically valid approach is by carrying out a noise impact study in one and the same area, which this group has previously done by creating an artificial highway (loudspeakers) and in the case of the current manuscript by creating artificial acoustic rivers, also with loudspeakers. I am skeptical about some of the reported results in previous studies on the impact of anthropogenic noise, but studies like the current one that have a valid control I do trust, even though some of the results surprised me. Note that the current study is not a simple repetition of the highway-study because water-noise is a completely different kind of noise from noise of motorized vehicles and wheels on asphalt. It is obviously also a type of noise that is evolutionary most relevant to animals. The reason other researchers have not embraced this type of approach is probably because most labs do not have access to remote natural areas with no risk of vandalism and theft of equipment. I have rarely seen a study that was carried out as meticulously as this one, with a control for practically every assessment as is laid out in a 89 (!!) page long supplementary. Many of the anthropogenic noise studies are disappointing because they stop short of the evolutionary relevant questions, whereas these questions are the starting point of this manuscript. Here the authors are describing how noise affects abundance and even foraging behavior of local animal communities. The

study has such detail that we even learn that –when controlled for abundance- birds are impacted on their ability to visually find prey (dummies) under noisy conditions. We also learn that up to certain noise levels, the frequency each bird uses can make it more resilient against noise (which by itself was never fully proven), but also that when noise levels get louder, frequency partitioning (if taken in evolutionary sense) no longer helps! All of this is new and very inspiring information because it leads us to the point where we can start to understand whether birds actually shape their calls as a function of ambient noise, which has been speculated about many times, without any adequate tests or data. Also we learn that bats near noise switch foraging strategy by hunting in more open areas. The current study provides very detailed quantifications of the impact of noise in dB on the local distribution of birds, even on species level, which we have never seen in other studies.

It is a shame that the manuscript is not a full-length manuscript because it is so full of information, yet it reads well and is easy to follow. I have no corrections or critical comments. On the bat-detections and the higher-frequency species being affected more by noise than low-frequency ones, could this possibly be a side effect of high-frequency bats being detected only when they are near the microphone, whereas low-frequency species passing over high will be detected more easily in both noise and no noise conditions?

I would like to state that –as a reviewer- I am weak on statistics but strong on acoustics. I can see that the acoustic side was also done meticulously, even down to details such as preventing spikes in the recording when going from one noise recording into the next and creating identical perceived noise levels in birds when shifting up the frequency by using octave band weighted recordings. Everything up to engineering standards. This manuscript stands out methodologically and also conceptually and I hope it will push away biologists from their narrow focus on anthropogenic noise effects to evolutionary more relevant noise sources and the responses of animals to them. If there is a reviewer who raised an issue that I missed I would be willing to cross-check whether I have been overly positive.

Reviewer #3 (Remarks to the Author):

I believe the idea to move back from anthropogenic noise impact to natural noise impact studies with methodological tools developed in urban studies is outstanding.

The phantom river is a well-suited experimental approach that is conducted in a proper way.

The mix of descriptive and experimental studies makes this a strong paper. The mix of data on birds and bats add beyond-single-taxon credibility, but also make it a long and sometimes less clear paper.

The findings for both birds and bats are well-supported and interesting.

The shifted river exposure treatment is a bit odd, and they probably could have better used another natural noise source with a higher-frequency distribution than river noise (i.e. cicada/grasshopper noise).

The novelty is not in the impact of noise on community composition or behavioural tendencies, as these phenomena have been reported for anthropogenic noise repeatedly, but are novel for natural noise effects in an experimental study like this.

I believe the paper can become more clear by explaining better what positive and negative control sites are as I found the maps in the main text and supplement difficult to understand. Why was there a biased distribution in control sites or positive control sites to one area and treatment sites in the other, and can the authors explain why this does not lead to a problematic confounding factor (area)?

REVIEWER COMMENTS – black

Author responses - Red

Reviewer #1 (Remarks to the Author):

This is a great piece of work, containing an impressive amount of field observations and experimental manipulations. The authors compared natural sites with varying levels of riverstream noise with sites where they broadcasted the same as well as upshifted noise through speaker arrays for two seasons. The results are extensive and very interesting and will stimulate a lot of future research.

Thank you for these kind comments!

I have only a few comments/questions regarding the methods and results, which I have added to the manuscript.

We have pasted comments from the annotated manuscript (only Reviewer 1) throughout (where relevant to other questions) to be able address them here, rather than in two separate documents.

All the analyses focus on aspects of the background noise, irrespective of their nature. My main question is why the authors did not provide details on comparisons between the control and experimental sites and only report patterns with noise level or frequency content in their main text. Perhaps it's hidden in the supplementary files, but I thought it was a bit odd to provide a lot of details on the sampling setup which was not reflected in the results.

This is a great question. We struggled with this ourselves, quite extensively. We created, as you are aware, specific treatments for some of the sites, while leaving others as controls. While this design seems prime for a categorical analysis (i.e. comparing experimental to control sites), we felt that such an analysis would only crudely capture the actual variation in the patterns we see since both experimental and control sites varied dramatically in sound levels and frequency measurements due to varying background environments. Control sites, for example, spanned a tremendous range of sound levels, from naturally quiet to extremely loud sites, meaning that these sites should not affect animals similarly if bats and birds are responding to either sound levels or frequency or both. In addition, each of our sites contained three sampling locations that were far enough apart to have markedly different acoustic environments. Thus, even experimental sites contained a large range in sound levels and frequency values. For these reasons, we did not feel that a categorical analysis would tell us anything about how animals respond to the characteristics of sound itself – something that only long-term, continuous monitoring of the acoustic environment was going to tell us.

Copy and pasted from comment 10 in annotated manuscript:

What appears to be missing from the main text are comparisons between the experimental treatment noise levels and the control noise levels. In other words, do you find the same patterns at control sites compared to treatment sites?

Please see answer to previous question

Comment 12:

I think the data in the figure very unconvincing, which makes me doubt about the type of models you used. The plotted regression lines hardly show a decline, the raw data seems very scattered, with no obvious pattern visible and the top and bottom part of the grey area around the line overlap across your range of noise levels. How can this be statistically significant?

I would take another close look at the model structure/assumptions and/or your figures.

The data in the figure (the question is referring to Figure 2A) are raw points between bird abundance and sound pressure levels. The line and confidence interval represent predicted values of abundance over the range of sound pressure levels **given mean values of all other variables in the model**. This last part is critical for the interpretation of these predicted values in relation to the plotted raw data points. We have now better highlighted this in the figure captions (Figures 2 and 3). We do not *necessarily* expect to be able to see any obvious visible pattern with abundance by sound level alone. The point of the mathematical model is to include many dimensions (variables) and allow maximum likelihood to estimate what the effects of are any **one** of those dimensions **given** the many other dimensions. Here we can only plot abundance with one other dimension – sound level. But, underlying the predicted model are 9 other fixed effects that are considered in the predicted model line and confidence intervals. Indeed the line isn't a sharp decline, but it is a 7% decrease every 12 dB, as explained in the text (please see response below). Generalized linear mixed effects models are the gold standard in statistical ecology, and are indeed appropriate here. The fact that the bottom of the shaded area on the left side of the plot overlaps with the top of the shaded area of the right side of the plot does not indicate anything about statistical significance, and further does not *necessarily* indicate that there is possibly a flat line between those two points within the shaded area. The top of the shaded area indicates the upper range of what the line might look like, while the bottom of the shaded area indicates the lower range of that same predicted estimate line, this does not mean that all possible combinations of lines exist within this shaded area.

I also have some concerns regarding the results plotted in Fig 2. The regression line and CI don't seem to match the reported effects in the text, so I would advise the authors to have an other careful look at the models or the figure.

Thank you for this concern. What may be confusing about the results reported in the text is that they are in percentages. Upon inspection of our figures, we find that the figures do in fact show similar results as reported in text. Figure 2A, for example, is reported as a 7.0% decline every 12 dB in the text. When we draw vertical lines at 30 and 42 dB (12 dB difference) and then draw horizontal lines where the trend line crosses these lines (See image below for example), we get two values on the y axis for predicted abundance. Using measuring tools in Inkscape, we get the value 5.94 and 5.52. Thus, the percentage difference across this 12 dB range = $(5.94 - 5.52) / 5.52 \times 100 = 7.43\%$

$5.52/5.94 = 0.0707$ or 7.1% (rounding up). Given such crude measuring methods, this is incredibly close to our reported value (7.0%) in the text.

Keep up the good work!

Thank you!

Copy and pasted comments from annotated manuscript:

Comments 5 and 7:

This study is highly debated. It's unclear whether these amphibians are sensitive to airborne sounds, the study lacks a good control and is based on a single replicate!

Ah, perhaps use this and/or the reef sound example instead of the Salamander study.

Thank you for this point. We were unaware of the debate surrounding the salamander study, and have replaced it, as you suggest, with the bats and rain noise study.

Comment 9:

This totally depends on species-specific perceptual processes. Birds can show very diverse audiograms in terms of shape and range, which can also differ from hearing thresholds in the presence of noise.

I would suggest to remove this statement and instead focus on the specific mechanism you aim to test... which to me appears to be masking

Thank you for this comment. We created our playback files using audiograms from 6 species that are found in our study area (see supplement), yet we take your point that hearing thresholds differ in noise (data that is scarce across bird species). We have now altered this sentence to read: "We created these files so that the average broadcast energy weighted by birds' hearing thresholds was the same (see supplement for details; Fig. 1C)."

Comment 11:

Quoted text: *"Our experimental design allowed us to explicitly test the effects of sound level separately from those of background spectra."*

I would disagree, given your lack of knowledge on the perceptual properties of species involved.... I also think the paper does not need this statement, so I suggest to remove it. You are correct in that we cannot know what the perceived loudness or pitch of the acoustic environment is, but that is not exactly what we are saying here. We are saying that due to our shifted noise treatments, we are able to estimate the effects of background frequency on birds and bats separately from sound pressure levels. Both frequency and sound level are acoustic characteristics of the environment and, as measured, do not depend on how they are being perceived. Thus, we are still able to independently test how bird abundance and bat activity differ along the range of those two variables.

Comment 13:

I think Nature Communications allows for some tables too..... I'd prefer the most important test summaries in the main text, not in the supplementary information

Thank you for this comment, and the desire to have more tables up front. One problem with this is that Nature Communications suggest only 4 figures/tables for the length of our short manuscript (for formatting purposes). So, we would only have room for 1 table. There is not one table from our many that clearly stands out above all of the others, so we have elected instead to put all tables in the supplement where they can be found together. However, we do report the most important test summaries in the main text.

Comment 14:

This is a very interesting observation. Could you plot that in a striking way?

Thank you for this comment. We have now plotted this as supplemental figure S14, and refer to it in the main text where you bring up this question.

Comment 15 referring to Fig 2C:

Now this is a much clearer pattern. It also suggest that the pattern in 2B is not significant... assuming the error bars are 95% CI, most of them overlap zero..... So no overall effect of spectral overlap on abundances.....

Figure 2B shows the overall pattern of all birds combined, while Figure 2C is showing individual species' estimates for those species that we had enough data to estimate individually. These

are quite different things. The fact that about half of the error bars overlap zero does not negate the possibility of a statistically significant effect in birds overall. Some birds are far more common for example, and can drive patterns more than others that are less common. Some birds that weren't common enough to model individually are also missing from Figure 2C.

Comment 16:

This is a review about noise and light pollution, not about cross-modal impacts. Better cite Halfwerk & Slabbekoorn 2015, BL or Halfwerk & van Oers 2020.

Thank you for pointing this out. We have now replaced Swaddle et al. 2015 with Halfwerk and Slabbekoorn 2015 as you suggest.

Comment 18:

both studies are about synthetic or anthropogenic noise I think. I don't know of any evidence of natural noise impacts?

This comment is referring to two references (below), one of which (16) used natural noise from wind blowing through rushes (along with other noise playbacks):

16. Schaub, A., Ostwald, J. & Siemers, B. M. Foraging bats avoid noise. *Journal of Experimental Biology* 211, 3174–3180 (2008).

17. Gomes, D. G. et al. Bats perceptually weight prey cues across sensory systems when hunting in noise. *Science* 353, 1277–1280 (2016).

Comment 22:

I don't understand this argument, please rephrase or elaborate

Thank you for asking us to clarify this argument. We have now done so in the main text. Please let us know if it is still unclear.

Comment 23:

I find the reference to specific statistics a bit all over the place. Sometimes you refer to tables, sometimes a few test statistics.....Please be consistent. And if you do report test statistics in the main text, please specify additional variables (N ...) and mention the type of test.

Throughout the text we report test statistics where that information within a sentence is supported by an individual model, and thus reporting is straightforward. Sometimes, however, we resort to pointing readers towards tables because our statement is referring to many models, and reporting would be cumbersome. For example, the statement "Individual-species models reveal consistently similar inferences (Table S6)." is pointing the reader to a table that is 6 pages long.

We appreciate the desire to have the sample size reported in the text. However, with mixed effects models the sample size is not straightforward. If we monitor 60 locations, nested within 20 sites, and monitor each of those locations such that we have thousands of observations due to repeated measures, do we have a sample size of 20 or the number of observations? Statistically, the "sample size" is treated to be somewhere in between those two numbers, but

figuring out exactly what this is – and more importantly what that means – is not straightforward like it with simpler statistical methods.

Comment 24:

Can you perhaps list the species here?

Thank you for this comment. We have now included the four species in the main text at this position.

Comment 25:

But this experimental finding is in contrast with the previous results that passive listeners are not effected by noise. How do you explain this discrepancy?

We do not state that passive listening bats are not affected by noise, but that they *weren't more likely* to be affected by noise than other bats, which was unexpected. Additionally, in this foraging test, we controlled for differences in bat abundance at the sites, meaning that these were behavioral shifts (not distributional ones). That is, bats might not have changed distributional activity, but did change foraging activity – and this change may indeed be the mechanism underlying these animals' resilience to noise exposure (as we state in the manuscript).

Comment 30:

What do the black dots inside the symbols signal?

Thank you for catching this important point. The black dots are spatially linking the sites in panel A with the spectrograms in panel B, to demonstrate exactly where those spectrograms came from. We have now clarified this in the figure legend.

Comment 33:

Please specify the error bars and grey are around the regression line

Thank you for point out that we neglected to specify these important points. We have now done so in the updated version of the manuscript.

Reviewer #2 (Remarks to the Author):

Major claims: natural noise impacts the distribution of animals along the noise gradient and can alter their foraging behavior. >Novelty: NATURAL noise, gradient-impact quantification, behavioral changes even cross-sensory (none of them demonstrated before or only poorly substantiated)

Importance to community: authors show evolutionary basis of adaptations in occurrence and implications on their vocalizations, a topic of speculation but up to now severely lacking in data. "Impact of natural acoustics on ecology" is a very general topic that few other studies contributed to as much as the current study.

This manuscript is a landmark contribution to the current knowledge of how noise shapes acoustic communities and how it affects local abundance of birds and bats. Much attention has

focused on the impact of anthropogenic noise, even with and without covid-lockdowns, or on presumed acoustic adaptations by birds to anthropogenic noise (some of which are still being questioned).

Many studies on the impact of anthropogenic noise have no valid control, simply because uninhabited areas with no roads (and noise) also tend to constitute more attractive habitats. The one and only scientifically valid approach is by carrying out a noise impact study in one and the same area, which this group has previously done by creating an artificial highway (loudspeakers) and in the case of the current manuscript by creating artificial acoustic rivers, also with loudspeakers. I am skeptical about some of the reported results in previous studies on the impact of anthropogenic noise, but studies like the current one that have a valid control I do trust, even though some of the results surprised me.

Note that the current study is not a simple repetition of the highway-study because water-noise is a completely different kind of noise from noise of motorized vehicles and wheels on asphalt. It is obviously also a type of noise that is evolutionary most relevant to animals. The reason other researchers have not embraced this type of approach is probably because most labs do not have access to remote natural areas with no risk of vandalism and theft of equipment. I have rarely seen a study that was carried out as meticulously as this one, with a control for practically every assessment as is laid out in a 89 (!!) page long supplementary. Many of the anthropogenic noise studies are disappointing because they stop short of the evolutionary relevant questions, whereas these questions are the starting point of this manuscript. Here the authors are describing how noise affects abundance and even foraging behavior of local animal communities. The study has such detail that we even learn that –when controlled for abundance- birds are impacted on their ability to visually find prey (dummies) under noisy conditions. We also learn that up to certain noise levels, the frequency each bird uses can make it more resilient against noise (which by itself was never fully proven), but also that when noise levels get louder, frequency partitioning (if taken in evolutionary sense) no longer helps! All of this is new and very inspiring information because it leads us to the point where we can start to understand whether birds actually shape their calls as a function of ambient noise, which has been speculated about many times, without any adequate tests or data. Also we learn that bats near noise switch foraging strategy by hunting in more open areas. The current study provides very detailed quantifications of the impact of noise in dB on the local distribution of birds, even on species level, which we have never seen in other studies.

It is a shame that the manuscript is not a full-length manuscript because it is so full of information, yet it reads well and is easy to follow. I have no corrections or critical comments.

Thank you for all of your kind words. We are elated that you appreciate the work that we put into this manuscript!

On the bat-detections and the higher-frequency species being affected more by noise than low-frequency ones, could this possibly be a side effect of high-frequency bats being detected only

when they are near the microphone, whereas low-frequency species passing over high will be detected more easily in both noise and no noise conditions?

You are correct that high-frequency bats are likely detected nearer the microphone however, as this analysis is comparing the relative differences in activity *within* species across acoustic environments, this is unlikely to be a problem. But perhaps we are missing something. Importantly, our various tests indicate that neither our automatic ultrasonic recording systems nor our analysis software show altered detection probabilities in noise.

I would like to state that –as a reviewer- I am weak on statistics but strong on acoustics. I can see that the acoustic side was also done meticulously, even down to details such as preventing spikes in the recording when going from one noise recording into the next and creating identical perceived noise levels in birds when shifting up the frequency by using octave band weighted recordings. Everything up to engineering standards. This manuscript stands out methodologically and also conceptually and I hope it will push away biologists from their narrow focus on anthropogenic noise effects to evolutionary more relevant noise sources and the responses of animals to them. If there is a reviewer who raised an issue that I missed I would be willing to cross-check whether I have been overly positive.

Again, thank you very much for all of your kind words!

Reviewer #3 (Remarks to the Author):

I believe the idea to move back from anthropogenic noise impact to natural noise impact studies with methodological tools developed in urban studies is outstanding.

The phantom river is a well-suited experimental approach that is conducted in a proper way.

The mix of descriptive and experimental studies makes this a strong paper. The mix of data on birds and bats add beyond-single-taxon credibility, but also make it a long and sometimes less clear paper.

The findings for both birds and bats are well-supported and interesting.

Thank you for this positive feedback!

The shifted river exposure treatment is a bit odd, and they probably could have better used another natural noise source with a higher-frequency distribution than river noise (i.e. cicada/grasshopper noise).

Thank you for this comment. We agree that it is indeed less natural than cicada noise – and to us often sounded like cicada noise. However, we were hoping to completely isolate the effects

of frequency from the temporal profile of the noise. If cicada noise led to differences in patterns of animal abundance or activity, we wouldn't be able to pin down whether or not those changes were due to differences in frequency or differences in temporal modulation profiles.

The novelty is not in the impact of noise on community composition or behavioural tendencies, as these phenomena have been reported for anthropogenic noise repeatedly, but are novel for natural noise effects in an experimental study like this.

I believe the paper can become more clear by explaining better what positive and negative control sites are as I found the maps in the main text and supplement difficult to understand. Why was there a biased distribution in control sites or positive control sites to one area and treatment sites in the other, and can the authors explain why this does not lead to a problematic confounding factor (area)?

We used 5 river drainages, 2 of which contained natural controls only (bottom-left of Fig 2A), and 3 of which contained a combination of natural controls and experimental sites (top-right of Fig 2A). At those three drainages we pseudo-randomly distributed treatment sites throughout, matching based on elevation and vegetation. That is, within three of our drainages (16 of our 20 sites) sites were well-balanced throughout the area. We were not allowed by land management agencies to experimentally manipulate the other two drainages (4 sites). Note in Fig 2A that all of these sites are actually quite close in proximity, we just zoom in to two areas for esthetic appeal. The figure in the supplement (Fig S1) was actually misleading, and outdated. When we started this experiment, we had considered the 4 sites from the two drainages to be louder whitewater sites – or 'positive controls', and the rest to be quieter streams – or 'negative controls'. However, as you can see in Fig 1A in the main text, this was not the case as sound levels were naturally distributed across the study. We have fixed Fig S1 to reflect this and updated the caption, and hope that it clears up any confusion.

This does not lead to a problematic confound by area because of the design. We analyzed our data based not on category (control, river noise, and shifted noise), but based on sound pressure level and frequency – which varied within each site (both by location and date). Thus, each of the 20 sites had three locations (60 total) which experienced a range of sound levels and frequencies over the course of the two-year study for the regression analysis. By using sites as random intercepts, we allow the mean number of birds/bats at each site to vary (in case there are any overall differences in sites / drainages). That is, the slopes of the relationship between bird abundance / bat activity and sound pressure level / background frequency will not change because of what is happening with latent variables (unforeseen differences within drainages) at a subset of sites. That is the power and beauty of random effects terms within regression analyses.

REVIEWER COMMENTS

Reviewer #1 (Remarks to the Author):

I thank the researchers for their detailed reply to my comments/suggestions. Re-reading the manuscript I was struck again by the amount of work and information. Really impressive.

I am however not fully convinced about the reasoning to pool control and experimental sites and focus on background noise level and spectra instead.

Effectively, that turns the data into an observational study and one could argue that all of the reported patterns reflect a confounding factor such as vegetation linked to noise level through size of the streams and associated humidity.

Is there any way to do justice to the experimental work? Even if you'd simply plot separate regression lines for the control vs experimental sites? I am aware you'll lose power if you'd added treatment as factor in addition to noise level or spectrum, let alone modelling their interaction, but simply plotting the data would at least show us that the patterns are the same and thus cannot be explained by confounding variables.

At least I am not the only one who got confused by the experimental design and mismatch in the result section, so I would urge the authors to reconsider this, even though I am convinced it will not affect the main messages from this otherwise great piece of work.

Reviewer #2 (Remarks to the Author):

The authors have sufficiently addressed my previous comments. I have no further comments / criticism.

Reviewer #3 (Remarks to the Author):

I believe the authors responded adequately to all comments of all reviewers. Looking forward to see this published.

Reviewer #1 (Remarks to the Author):

I thank the researchers for their detailed reply to my comments/suggestions. Re-reading the manuscript I was struck again by the amount of work and information. Really impressive.

Thank you!

I am however not fully convinced about the reasoning to pool control and experimental sites and focus on background noise level and spectra instead.

Effectively, that turns the data into an observational study and one could argue that all of the reported patterns reflect a confounding factor such as vegetation linked to noise level through size of the streams and associated humidity. Is there any way to do justice to the experimental work? Even if you'd simply plot separate regression lines for the control vs experimental sites? I am aware you'll lose power if you'd added treatment as factor in addition to noise level or spectrum, let alone modelling their interaction, but simply plotting the data would at least show us that the patterns are the same and thus cannot be explained by confounding variables.

At least I am not the only one who got confused by the experimental design and mismatch in the result section, so I would urge the authors to reconsider this, even though I am convinced it will not affect the main messages from this otherwise great piece of work.

Thank you for this point. We find this point highly valuable to the understanding of our results and the dissemination of this paper. Thus, we have incorporated most of this response here within the supplementary file, including expanding Tables S1 and S5.

While we appreciate the point, we do not entirely agree that the data are turned into an observational study by analyzing a continuum of a variable, rather than analyzing it categorically. We also do not agree that it necessarily takes away from the experiment. Figure S6 shows nicely that all sites were strongly overlapping in sound levels (pre noise exposure), with the majority of the sound levels between 35 and 60 dB. When the experiment is turned "on", the majority of the energy in those experimental sites is now between 60 and 80 dB (right panel in Fig S6). Thus, nearly all of the high sound level sites came from the experiment. If there were confounding latent variables at the sites, we wouldn't expect to see such a strong signature of sound level in our analysis, since the streams that were experimentally made to have high sound levels originally had lower levels (Fig S6), thus the confounds of having larger streams and more humidity would not track linearly with sound levels – hence the power of the experiment. We do recognize that it is possible in control sites that the effects of sound level are correlated with a latent variable related to stream size, but those data are so much more sparse at the high end than the experimental data.

Additional confounding variables that almost certainly share some correlation to stream size and flow (and thus can absorb some of this variation in the model) are date (day of year), a quadratic version of date, riparian vegetation, and elevation – all of which were in our models, which help separate the effects of noise measures from other confounds. Thus, with this

experimental design and analysis, we find it extremely unlikely that the effects would be due to a stream size confound.

In order to alleviate these concerns, however, we have quantitatively assessed bat and bird activity and abundance (respectively) in experimental sites only (see below). Yet, since we feel that these treatments are mostly human constructs and can be misleading, we prefer to keep this analysis in the supplemental information (with some caveats), rather than in the main body of the manuscript. The reasoning is as follows. We broadcast noise at some locations and not others. While this would seem, on the face of it, that there are clear designations of treatments, there are not. We chose locations that were some distance from each noise playback area to create a continuum of sound pressure levels and background frequencies. Thus, at many locations you can hear the river or shifted river broadcasted files, but nothing near the intensities at the playback locations themselves. So it appears that we have created continua of treatments – or *realized* treatments. These realized treatments could get some arbitrary value of weighted treatment that we come up with ourselves (e.g. 0.5river, 1river, 0.5shifted, 1 shifted) to incorporate the differences in intensity and spectrum. However, we should instead quantify the acoustic environment and ask specifically what different components of noise (i.e. sound levels and frequency) do to drive wildlife populations and behavior, which also happen to help with the mechanistic understanding of the system. We would argue that a categorical analysis of a gradient is almost always going to be a more crude approach than a continuous one, and should only be used when continuous data aren't available or when categories actually do not differ in the values that one would measure continuously.

The quantitative approach:

In an attempt to add treatment interactions to sound level estimates, we run into a problem of high multicollinearity, or non-identifiability of the parameters. That is, the slope (estimate) of sound levels for control data is highly predictive of (and thus also predicted by) the slopes for the interaction of the treatments by sound level – which might suggest that these slopes are nearly identical and the treatment term isn't doing anything. In this situation, calculations regarding these predictors cannot be trusted. We ran into this issue with a fully-specified model (original models with treatment interactions added) and within a simplified model (only sound level and treatments + interactions). Thus, we cannot accurately estimate the differences between these slopes given such high variance inflation factor (VIF) scores for these parameters (Bird abundance: VIF = 23.7 for sound level by treatment interaction in full model; Bat activity: VIF = 19.2 for sound level by treatment interaction in full model; both of these scores are considered high by any assessment).

Thus, plotting model predicted lines are not necessarily going to reflect accurate estimates. For this reason, it seems the only viable way to check inferences from experimental data is to create models that only include experimental data.

At the top of Table S1, the first global bird model presented includes estimates for sound pressure level of -0.067 with a standard error of 0.018. The same model including only experimental sites (now included at the bottom of Table S1) produces an estimate of -0.052 with

a standard error of 0.019. Both of these estimates return p values below 0.01 and are overlapping estimates considering the standard errors presented here.

At the top of Table S5, the first global bat model presented includes estimates for sound pressure level of -0.082 with a standard error of 0.018. The same model including only experimental sites (now included at the bottom of Table S5) produces an estimate of -0.097 with a standard error of 0.019. Both of these estimates return p values below 0.001 and are overlapping estimates considering the standard errors presented here.

We hope that the arguments presented here, along with the additional analyses of experimental data convince you, and future readers, that our experiment is not the result of confounding latent variables associated with streams. We look forward to your response.

Reviewer #2 (Remarks to the Author):

The authors have sufficiently addressed my previous comments. I have no further comments / criticism.

Reviewer #3 (Remarks to the Author):

I believe the authors responded adequately to all comments of all reviewers. Looking forward to see this published.

REVIEWERS' COMMENTS

Reviewer #1 (Remarks to the Author):

Great reply. All is clear now. No further comments

Reviewer #1 (Remarks to the Author):

Great reply. All is clear now. No further comments

Thank you for improving this manuscript.